# Protective Effect of Tuna Bioactive Peptide on Dextran Sulfate Sodium-Induced Colitis in Mice

**DOI:** 10.3390/md19030127

**Published:** 2021-02-26

**Authors:** Xing-Wei Xiang, Xiao-Ling Zhou, Rui Wang, Cong-Han Shu, Yu-Fang Zhou, Xiao-Guo Ying, Bin Zheng

**Affiliations:** 1College of Food Science and Technology, Zhejiang University of Technology, Hangzhou 310014, Zhejiang, China; xxw11086@zjut.edu.cn (X.-W.X.); healing0852@gmail.com (R.W.); 2Key Laboratory of Marine Biological Resources Innovation and Development of Zhejiang Province, Hangzhou 310014, Zhejiang, China; 3Food and Pharmacy College, Zhejiang Ocean University, Zhoushan 316000, Zhejiang, China; zhouxiaoling69@163.com (X.-L.Z.); shuconghan@163.com (C.-H.S.); yingxiaoguo@zjou.edu.cn (X.-G.Y.); 4Zhejiang Marine Development Research Institute, Zhoushan 316000, Zhejiang, China

**Keywords:** low molecular weight peptides, ulcerative colitis, inflammation, intestinal barrier, intestinal flora, short-chain fatty acid

## Abstract

Bioactive peptides isolated from marine organisms have shown to have potential anti-inflammatory effects. This study aimed to investigate the intestinal protection effect of low molecular peptides (Mw < 1 kDa) produced through enzymatic hydrolysis of tuna processing waste (tuna bioactive peptides (TBP)) on dextran sulfate sodium (DSS)-induced ulcerative colitis (UC) in BALB/c mice. Here, we randomly divided twenty-four male BALB/c mice into four groups: (i) normal (untreated), (ii) DSS-induced model colitis, (iii) low dose TBP+DSS-treated (200 mg/kg/d), and (iv) high dose TBP+DSS-treated groups (500 mg/kg/d). The results showed that TBP significantly reduced mice weight loss and improved morphological and pathological characteristics of colon tissues. In addition, it increased the activities of antioxidant enzymes (SOD and GSH-Px) and decreased inflammatory factors (LPS, IL-6, and TNF-α) expression. TBP increased the gene expression levels of some tight junction (TJ) proteins. Moreover, TBP increased the short-chain fatty acids (SCFAs) levels and the diversity and imbalance of intestinal flora. Therefore, TBP plays some protective roles in the intestinal tract by enhancing antioxidant and anti-inflammatory abilities of the body, improving the intestinal barrier and metabolic abnormalities, and adjusting intestinal flora imbalance.

## 1. Introduction

Ulcerative colitis (UC) is a disorder characterized by non-specific chronic intestinal inflammation, and the World Health Organization (WHO) has classified it as a refractory disease [1]. Its prevalence has gradually increased globally, and that of China has risen to 11.6/10^6^ [2]. The disease has no prominent age characteristics. Patients with low resistance are more prone to recurrent abdominal pain, diarrhea, and other clinical symptoms, which seriously affect their everyday social life [3]. Researchers generally believe that its etiology correlates with the complex interaction between environmental factors, immune dysfunction, oxidative stress, and intestinal flora disturbance [4]. Furthermore, in these patients, the intestinal flora disturbance causes a series of adverse effects on the body [5]. These include reduced gut ability to resist invasion by foreign pathogens, damage to intestinal tissues, and an aberrant immune response [6,7]. Recently, numerous studies have reported that dietary factors can indirectly affect the UC pathogenesis process by directly acting on the host or by regulating the composition or function of intestinal microorganisms [8].

Bioactive peptides are specific protein fragments that positively affect body functions and have attracted progressive attention in preventing and controlling diseases [9]. Usually, they are inactive in protein sequences but can function after being released through enzymatic hydrolysis or microbial fermentation [10]. They have relatively small molecular weight and are absorbed easily across membranes. Apart from nutritional benefits, bioactive peptides have several physiological regulatory effects, like lipid lowering, antibacterial, and immune regulation [11]. Moreover, previous literature illustrates that dietary peptides derived from therapeutic food proteins can be delivered to colon epithelial cells to reduce inflammation and modulate pro-inflammatory signaling events [12]. Azuma [13] showed that the collagen peptide in fish scales played an anti-inflammatory role in the dextran sulfate sodium (DSS)-induced acute UC mice model. Liu [14] showed that bioactive peptides from egg whites contribute to maintaining healthy homeostasis and intestinal microflora of colitis hosts. Jin [15] illustrated the inhibitory effect of a new peptide from Pacific oysters (*Crassostrea gigas*) enzymatic hydrolysate on colitis induced by DSS in mice.

Tuna is a type of marine organism with high economic value, which contains many nutrients that are beneficial to the human body and is a high-quality protein resource [16]. Unutilized parts (tuna skeleton, dark muscle, fish skin, fish scales, etc.) generated during tuna processing are discarded, which account for about 50% to 70% of the original materials, hence, the problem of insufficient utilization of resources [17]. Enzymatic hydrolysis of these protein-rich byproducts is conducive to forming peptides with nutritional and functional properties [18]. Studies have reported that the derived peptides extracted from different parts of tuna have rich functions, like antioxidative [19], anti-obesity [20], bacteriostat [21], and inhibition of cancer cell proliferation [13]. Some studies have shown that some peptides isolated from tuna sources exert anti-inflammatory effects in vivo and in vitro. Cheng [22] showed that the small molecular peptides isolated and purified from the enzymatic hydrolysis of tuna boiled juice had great anti-inflammatory activity. Zhang [23] demonstrated that antioxidant peptides extracted from tuna bone protein enhanced the inflammatory response of necrotizing colitis caused by decreasing reactive oxygen species (ROS) and inhibiting the NF-κB pathway. Wang [24] illustrated that enzymatically hydrolyzed peptides from Skipjack (*Katsuwonus pelamis*) had a good regulating effect on inflammation and intestinal flora of UC model mice induced by DSS.

Skipjack is a kind of tuna species with a large fishing amount and has excellent market potential. In this study, the low molecular peptides (Mw < 1 kDa) (tuna bioactive peptides (TBP)) were prepared through enzymatic hydrolysis and membrane separation of skipjack processing waste to establish the effects of DSS on UC mice. Furthermore, our study would provide a theoretical basis and reference for the high-value utilization of tuna byproducts and the development of Marine biology-assisted treatment for intestinal inflammation.

## 2. Results

### 2.1. Characterization of TBP

As demonstrated in Table 1, we determined the molecular weight distribution of TBP through high-performance liquid chromatography (HPLC). TBP is mainly composed of molecules in the range of 180–500 Da (48.71%) and 500–1000 Da (43.23%). The present study results indicated that TBP was nearly a small molecule after enzymolysis and membrane separation. The amino acid analysis revealed that TBP contained seventeen kinds of amino acids, among which glutamate, aspartate, leucine, alanine, and lysine were the main components (Table 2). The essential amino acids (Leucine, Lysine, Valine, Histidine, Isoleucine, Isoleucine, Threonine. Phenylalanine and Methionine) and branched-chain amino acids (Valine, Valine, Isoleucine) accounted for 26.84% and 11.73% of the total amino acids.

### 2.2. Effects of TBP on Body Weight and DAI Score of DSS-Induced Colitis Mice

In this study, we established four groups of mice with different treatment, including Normal, DSS-induced model colitis (DSS), low dose of TBP (L-TBP), and the high dose of TBP (H-TBP) groups. The animal experiment process is shown in Figure 1A. As illustrated in Figure 1B, we observed a body weight change in each group of mice. During the last week of the experiment, the mice in the Normal group continued to gain weight. The DSS and TBP groups began to lose weight on the third day after DSS induction. Notably, we observed a reduction in weight loss in the TBP group than in the DSS group. The mice in the DSS group had mild diarrhea symptoms, such as reduced dietary water intake and activity and poor hair color, which significantly increased the score evaluation of the disease activity index (DAI). This suggests that the mouse colitis model was successfully constructed. After TBP treatment, we observed that the physiological condition of mice was improved to some extent but still existed, and the DAI score was lower than that of the DSS group.

### 2.3. Effects of TBP on Pathological Changes of Colon Tissues in DSS-Induced Colitis Mice

Figure 2 demonstrates the colon tissues of mice in each group. In the Normal group, the colonic epithelium had a clear structure. The glands were neatly arranged, the crypt and mucosa were intact, rich in goblet cells, and no apparent inflammatory cell infiltration was observed (Figure 2A). Scanning electron microscopy (SEM) and transmission electron microscopy (TEM) showed that the colonic epithelial microvilli of normal mice were closely arranged and of the same shape and size (Figure 2B,C). The colon tissues of mice induced by DSS were disordered, crypt destruction was severe, the number of goblet cells in the intestinal epithelium was significantly reduced, and many inflammatory cells were infiltrated in the mucosal and submucosal layers (Figure 2A). The mice colonic epithelial microvilli were enlarged, damaged, collapsed, and atrophic (Figure 2B,C). The colonic condition of the H-TBP group was better than that of the DSS group. Many goblet cells were retained, visible crypt structure appeared, and inflammatory cell infiltration significantly reduced (Figure 2A). The colonic epithelial microvilli of H-TBP group mice became denser, and the collapsing atrophy decreased (Figure 2B,C). These results suggested that TBP can slowed down the process of intestinal injury induced by DSS.

### 2.4. Effects of TBP on Cytokines of Colon Tissues in DSS-Induced Colitis Mice

Figure 3 shows the cytokines detection in colon tissue of mice in each group. Compared with the Normal group, the levels of inflammatory cytokines (IL-6 and TNF-α) in colon tissues of the DSS group were significantly increased (P < 0.05). On the other hand, H-TBP and L-TBP treatment significantly reduced the levels of these inflammatory cytokines, which was substantially different from that in the DSS group (P < 0.05) (Figure 3A,B). We further detected the SOD and GSH-Px activities in colon tissues and established that the activities of these antioxidant enzymes were significantly decreased in the DSS group (*p* < 0.05). Compared with the DSS group, the activities of SOD and GSH-Px were increased considerably after H-TBP treatment (*p* < 0.05) (Figure 3C,D). Furthermore, TBP effectively reduced intestinal permeability in mice. The LPS and DAO levels in colon tissues of the DSS group were significantly increased (*p* < 0.05). Compared with the DSS group, the LPS and DAO levels in L-TBP and H-TBP groups were significantly decreased (*p* < 0.05) (Figure 3E,F). 

### 2.5. Effect of TBP on Related Gene in Colon Tissues in DSS-Induced Colitis Mice

Like the results of colonic cytokines, the gene expression of TNF-α and IL-6 in colon tissues of the DSS group was also significantly increased (*p* < 0.05). After TBP treatment, the gene expression of TNF-α and IL-6 in colon tissue were decreased and approached those of the Normal group (Figure 4A,B). In addition, as illustrated in Figure 4C,E, DSS-induced mice significantly reduced the gene expression of tight junction (TJ) proteins such as ZO-1, Occludin, and Claudin (*p* < 0.05) in colon tissues. Compared with the DSS group, TBP treatment could increase the gene expression of TJ protein; however, H-TBP was significantly different. These results suggests that TBP therapy may reduce inflammation and heal intestinal damage.

### 2.6. Effects of TBP on the Gut Microbiota in DSS-Induced Colitis Mice

Here, we used Venn map, α-diversity index, and PCA map to evaluate the effects of H-TBP on gut microbiota structure. The Venn diagram illustrated that the number of OTUs shared by the groups was 160. The number of unique OTUs in the Normal group, DSS, and H-TBP groups were 105, 121, and 126, respectively. However, as shown in Figure 5A,B, the number of OTUs in the DSS group was the lowest. DSS group demonstrated that DSS reduced the α-diversity indices (Chao1, Simpson, and Shannon indices) of the gut microbiota compared to the Normal and H-TBP groups. As illustrated in Figure 5C–E, TBP treatment could inhibit the decrease of α-diversity in DSS-induced mice. PCA analysis showed that the gut microbiota obtained in the DSS group was significantly separated from the other two groups, and the H-TBP group was close to the Normal group (Figure 5F). These results are indicating that different treatments changed the structure of the gut microbiota of the mice. 

Furthermore, we analyzed gut microbiota changes at the phylum and genus levels in the three mice groups. As demonstrated in Figure 6A, *Bacteroidetes* and *Firmicutes* are the main flora at the phylum level in the gut. Compared with the Normal group, *Bacteroidetes* increased, and *Firmicutes* decreased in the DSS-induced mice. In contrast, after H-TBP treatment, *Bacteroidetes* decreased, whereas *Firmicutes* increased. At the genus level, the abundance ranking of *Bacteroides, Lactobacillus, Alistipes,* and *Lachnospiraceae NK4A136* groups in the Normal group was from one to four, respectively. The abundance ranking of one to four of the DSS group was *Alistipes, Lachnospiraceae NK4A136, Lactobacillus,* and *Bacteroides.* Compared with the Normal group, the abundance of *Bacteroides* in the DSS group was significantly decreased, whereas that of *Alistipes* was increased considerably. As illustrated in Figure 6B, the abundance ranking in the H-TBP group, from one to four, was *Bacteroides, Lachnospiraceae NK4A136, Alistipes,* and *Lactobacillus*, respectively. Moreover, the results showed that six bacteria at the genus level (four increased and two decreased) were significantly altered in the H-TBP group in comparison to the Normal group (*p* < 0.05) (Figure 6C). This includes the *Oscillibacter*, *Desulfovibrio*, *Rikenellaceae RC9 gut group*, etc. Compared with the DSS group, 13 bacteria were significantly changed at the genus level in the H-TBP group (nine decreased and five increased) (*p* < 0.05) (Figure 6D). This includes the *Alistipes*, *Oscillibacter*, *Rikenellaceae RC9 gut group*, etc. These results suggested that the TBP treatment could regulate gut microflora.

### 2.7. Effects of TBP on Short Chain Fatty Acids (SCFAs) in DSS-Induced Colitis Mice

Intestinal microorganisms have many effects on the host by mediating microbial products, among which SCFAs are related to IBD pathogenesis. Furthermore, we analyzed the changes of SCFAs in intestinal contents of the Normal, DSS, and H-TBP groups (Figure 7). As shown in Figure 7A–F, the SCFAs contents that were significantly decreased in the DSS group (*p* < 0.05) include acetic acid, propionic acid, butyric acid, isobutyric acid, valeric acid, and isovaleric acid. The difference is that the content of caproic acid increased significantly (*p* < 0.05) (Figure 7G). Compared with the DSS group, H-TBP treatment significantly increased the range of these SCFAs in the intestinal tract of DSS-induced mice (*p* < 0.05) (Figure 7A–G). The results suggested that the TBP treatment may help regulate microbial metabolites in the gut.

In this study, we analyzed the relationship between the 30 most abundant genera and SCFAs, to investigate whether the changes in the SCFAs in mice correlated with enteric gut microbiota. The results showed that acetic acid, propionic acid, isobutyric acid, butyric acid, iso-valeric acid, valeric acid and *Bacteroides*, *Oscillibacter*, *Odoribacter*, *Ruminococcaceae series* and *Lachnospiraceae UCG-006*, *Lachnoclostridium* and *Butyricicoccus* were positively correlated to different degrees, while negatively correlated to *Alistipes*, *Rikenella*, *Lachnospiraceae NK4A136 Group*, *Rikenellaceae RC9 Gut Group*, *Alloprevotella*, and other bacteria (Figure 8).

## 3. Discussion

Bioactive peptides play an essential role in human immune function and metabolic activities [25]. Currently, enzymatic hydrolysis is an effective and safe method of obtaining bioactive peptides [26]. Membrane separation can further isolate high-value molecules from hydrolyzed protein by-products to improve their functional properties [27]. Previous research indicated that low molecular weight polypeptides could be readily absorbed and utilized in intact form, and their biological activities could be preserved during their gastrointestinal digestion [28]. This study prepared low-molecular bioactive peptides from skipjack processing waste through enzymatic hydrolysis and membrane separation. The results showed that the molecular weight of TBP was less than one Kilodalton, and it was rich in various amino acids. UC is a complicated disease, and drug treatment often has some severe side effects [29]. DSS-induced mice are used to establish a non-specific UC model with symptoms and pathological changes similar to human UC. They mainly manifest as intestinal epithelial cell replacement, imbalance intestinal physical barrier, dysfunction intestinal oxidative damage, intestinal mucosa inflammation, and intestinal microorganism disorder [30]. Therefore, we assessed the TBP effect and mechanism on DSS-induced colitis mice.

In this study, typical clinical symptoms of colitis and pathological changes in the intestinal tract were produced in mice induced using 3.5% DSS (*w/v*) [31]. These included poor diet, weight loss, and bloody stool. In the DSS group, we observed colonic tissue ulceration and microvilli damage in the colon epithelium of mice. Intestinal mucosal barrier abnormalities are a key feature of individuals with colitis [32]. H-TBP treatment inhibited weight loss, reduced DAI score, and improved the damage of intestinal epithelial barrier in mice. The LPS and DAO levels, which represent the severity of the intestinal injury, began to decrease in the TBP group. In addition, further experiments demonstrated that TBP increased the gene expression level of TJ protein in colon tissue. These results propose that TBP treatment is beneficial for restoring the intestinal barrier in colitis. This could be linked to the amino acid composition of TBP. Different studies have reported that glutamate and aspartate provide energy to mammalian intestinal epithelial cells through mitochondrial oxidation, regulating the AMPK mTOR activity pathway, promoting intestinal epithelial cell proliferation, and playing an essential role in maintaining integrity of intestinal mucosa [33,34]. Leucine provides energy for nutrient transfer and intracellular protein transformation to promote intestinal development, affecting intestinal barrier function [35]. Arginine is a conditioned, essential amino acid that plays a significant role in cell physiological activities. Studies have shown that it can protect cells from apoptosis caused by LPS-induced oxidative damage [36].

The intestinal epithelial barrier improves the inflammatory response by preventing local inflammation from becoming systemic [37]. It is reported that inflammatory cytokines such as IL-6 and TNF-α are critical signaling molecules in the inflammatory response of the body, and their expression level is positively correlated with the severity of UC. Their increase causes the abnormal distribution of Occludin protein, making it unable to localize and function on TJ, causing the intestinal homeostasis instability after the breakage of TJ [38]. TBP treatment reduced the levels of IL-6 and TNF-α inflammatory cytokines in mice colon tissues induced by DSS, and their mRNA expression levels were close to that of the normal level. Excessive production of reactive oxygen species is the critical factor to cause tissue damage and ulcer formation during UC pathogenesis [39]. The increase of SOD and GSH-Px enzyme activity can stabilize cell membranes and protect colon tissue [40]. This experiment showed that after DSS induction, TBP intervention could improve SOD and GSH-Px activity in colon tissues. This finding is consistent with previous studies showing that peptides extracted from tuna have better antioxidant effects [41]. These results propose that TBP can inhibit intestinal inflammation, enhance antioxidant capacity, and repair epithelial barrier, thus alleviating colitis.

It has been reported that peptides can alter colonic microbes, and bacteria can modify peptides in the gut. Regulating intestinal microbiota is considered to be an effective method for the treatment of colitis. Intestinal microorganisms have established an asymmetric relationship with the host through long-term evolution and play a significant role in the development and maturation of the host immune system [42]. The normal intestinal flora is rich in composition and diverse in variety of which 99% of these strains are mainly *Firmicutes* and *Bacteroides* [43]. In a delicate balance, beneficial bacteria and conditional pathogens maintain usual in vivo metabolism and immune functions [44]. Several studies have shown that imbalanced intestinal flora can directly invade intestinal epithelial cells to induce inflammation, and its metabolite endotoxin can also trigger intestinal inflammatory responses [45]. UC has obvious bacterial flora imbalance, mainly manifested in changes in the type, number, and function of intestinal flora. The experimental results showed that TBP treatment increased the diversity index of intestinal microbial, but it was not significantly different from the DSS group, which had certain limitations. Some reports have shown a decrease in the *Bacteroidetes/Firmicutes* ratio in the gut of patients with colitis [46]. In mice intestines, TBP treatment increased *Firmicutes* and decreased *Bacteroidetes*. *Bacteroides* are one of the dominant bacteria in the intestine, and their existence can help reduce inflammation [47]. TBP treatment increased *Bacteroides* that were decreased after DSS induction. These results have shown that TBP plays a crucial role in maintaining intestinal flora homeostasis.

Moreover, TBP could exert an anti-colitis effect through other pathways, like microbial metabolism. SCFAs are well-known signaling molecules for regulating diet, gut microbes, and hosts. SCFAs, including acetic acid, propionic acid, butyric acid, isobutyric acid, and hexic acid, are produced by microorganisms fermenting undigested carbohydrates [48]. They affect chronic diseases through various mechanisms and play an essential role in maintaining the metabolic stability of colon epithelial cells, protecting the colon from external injury, and potentially impacting colon disease [49]. SCFAs as the primary energy substrate can also play an anti-inflammatory and anti-cancer effect [50]. The changes in gut microbiota have a significant impact on SCFAs in the intestine. Our results showed that DSS reduced SCFAs in the gut, but TBP treatment increased them. Correlation analysis showed that there were different associations between the gut microbiota and the SCFAs. For instance, *Ruminococcaceaes* are considered to be anti-inflammatory factors since they produce SCFA, particularly butyric acid [51]. However, its abundance was increased after TBP treatment. Therefore, the beneficial effects of TBP were mediated not only by enhanced intestinal barrier function, but also linked with the gut microbiota and its metabolism.

## 4. Materials and Methods

### 4.1. Chemicals

Tuna was purchased from Zhejiang Rongchuang Food Co. LTD (Zhoushan, China). DSS (MW:36000) from MP Biomedicals (San Mateo, CA, USA). ELISA kits for TNF-α, IL-6 and LPS were purchased from Nanjing Jiancheng Institute of Biological Engineering (Nanjing, China) and Wuhan Boster Company (Wuhan, China), respectively. DAO, GSH-PX, and SOD detection kits were purchased from Wuhan Huamei Biology Co., LTD (Wuhan, China). Trizol Reagent, the cDNA Reverse transcription Kit, and the TB Green^TM^ Premix Ex TapTM Fluorescent Quantitative Kit were purchased from TaKaRa (Dalian, China). Fecal genomic DNA extraction kits were obtained from TianGen Biotechnology (Beijing, China). All other chemical reagents are commercially available analytical grade.

### 4.2. Preparation of TBP 

Briefly, we collected the Skipjack processing wastes and washed them dozens of times in distilled water to remove the remaining blood and fat. Then, we mixed Skipjack processing wastes with sufficient phosphate-buffered saline (PBS) for homogenization and attained a pH of 7.4. After adding pepsin (2500 U/g), the homogenate was hydrolyzed at pH 3.0, 37 °C for 4 h. Subsequently, to inactivate pepsin, 0.1 M NaOH solution was used to adjust the pH value of the homogenate to 8.0. Next, we added animal protease (2500 U/g) to the homogenate and hydrolyzed it again for 4 h. After terminating the enzymatic hydrolysis reaction, precipitation was performed through centrifugation, and the enzymatic hydrolysate was obtained. The ultrafiltration device with a retention of 1 kDa was used to treat the enzymatic hydrolysate and collected the small molecular weight (< 1 kDa) peptide components of the processed tuna wastes, concentrate, and freeze-drying. HPLC (Agilent Technologies, Inc., CA, USA) equipped with TSK gel G2000 SWXL analytical column (7.8 × 300 mm, 5 µm, TOSOH, Tokyo, Japan) was used to determine the molecular weight distribution of TBP. Its amino acid composition was determined using the Hitachi 835-50 amino acid analyzer (Hitachi, Tokyo, Japan), methods as previously reported.

### 4.3. Animal Experiment Design

We procured male BALB/C mice aged five weeks from the Zhejiang Academy of Medical Sciences. Upon arrival, the mice were placed in a controlled environment (23 ± 2 °C, for a 12 h light/dark cycle), and provided with a standard basic diet. After the 5-day adaptation period, all mice were randomly divided into four groups of six mice each: Normal, DSS-induced model colitis (DSS), low dose of TBP (L-TBP), and the high dose of TBP (H-TBP) groups. Notably, all experimental protocols for animals were formulated per the National Institutes of Health standards and approved by the Ethics Committee of Zhejiang Ocean University (2019003). All necessary measures were used to reduce animal suffering during experimental procedures.

Based on our previous laboratory experiment, the L-TBP and H-TBP groups were designed to receive 200 mg/kg and 500 mg/kg TBP gavage, respectively, once daily until the end of the experiment. The Normal and the DSS groups were given a similar amount of pure water. In the last week of the investigation, DSS (3.5% *v/w*) was added to the drinking water of the DSS, L-TBP, and H-TBP groups to induce experimental UC in mice (Figure 1A). At the end of the experiment, all mice were euthanized, and the colon tissues and cecal contents were collected and stored at −80 °C until further analysis.

### 4.4. DAI Score Evaluation

We used the DAI score to assess the severity of colitis in each group. After establishing the DSS model, we conducted the comprehensive score daily following the weight change of mice, including fecal viscosity and fecal hematochezia. The specific criteria were shown in Table 3.

### 4.5. Pathological Evaluation

After we separated the colonic tissues of mice in each group, some fresh tissues were fixed in 4% paraformaldehyde solution for 48 h and embedded in paraffin. Then, paraffin sections were prepared successively through block repair, dehydration, transparency, waxing, embedding, section, etc., and were stained with hematoxylin and eosin (H&E) solution, sealed with neutral gum, and observed under a microscope. Subsequently, we evaluated the degree of edema, superficial necrosis, and infiltration of granulocytes in colon tissues.

The partial colon tissue was carefully washed with PBS, fixed in 2.5% glutaraldehyde at 4 °C. After dehydration under different alcohol concentrations, the tissue block was dried at a critical point, pasted onto a short rod, coated with gold, and then observed under a scanning electron microscope (SEM). The other part of the treated colon tissue was impregnated with epoxy resin, made into ultrathin sections, and stained using uranium acetate and aluminum citrate for TEM observation.

### 4.6. Colon Tissues Biochemical Parameter Analysis

A section of colon tissue was taken. Normal saline was added at a mass volume ratio of 1:9, the tissue was ground using a homogenizer, and the supernatant was collected through centrifugation, that is, 10% of the colon tissue homogenate. Then, we measured the activity of DAO, SOD, and GSH-PX in colon tissue homogenate using a kit. Lastly, we determined TNF-α, IL-6, and LPS contents in colon tissue homogenate using an ELISA kit. Three biological replicates were performed for each treatment.

### 4.7. Real-Time Polymerase Chain Reaction (RT-PCR)

We extracted total RNA from colon tissues using the Trizol reagent and converted it into complementary DNA (cDNA), following the instructions from the manufacturer. cDNA is produced through RNA reverse transcription using the Prime Script™ Reverse Transcription Kit. Then, we performed quantitative PCR using SYBR Green PCR Core Reagent in ABI Step One Plus Real-Time PCR system. The procedures were performed as follows: Stage 1: Pre-denaturation at 95 °C for 30 s, one cycle; Stage 2: PCR, at 95 °C for 5 s and 60 °C for 34 s, 40 cycles; and Stage 3: at 95 °C for 30 s and 60 °C for 1 min, one cycle. All qPCR reactions were performed for each RNA sample. The gene expression levels were calculated using the 2−ΔΔCt method. β-actin was used as the internal control. Table 4 shows the primers used in this study.

### 4.8. Gut Microbiota Analyses

We extracted total bacterial DNA from samples of cecal contents using fecal genomic DNA extraction kits, following the instructions from the manufacturer. Then, V3–V4 regions of the bacterial 16S rRNA were amplified using the Phusion High-Fidelity PCR Master Mix with a specific primer barcode. After the purification of PCR products, a sequencing library was generated to assess the quality of the library. Finally, the library was sequenced using the MiSeq sequencing system (Illumina, San Diego, CA, USA), and the quality control program was read single-ended to acquire clean reads. Next, we used the Uparse software to allocate reads with a similarity of 97% to the same OTUs and further annotate them according to the SILVA database. The QIIME (Version 2) software was used to calculate the α-diversity index, including Chao, Simpson, and Shannon. Consequently, we used the STAMP software (Version 2.1.3) to plot PCA and compared the different flora of each group. Lastly, the Spearman coefficient was used to analyze the association between microbiota and SCFAs.

### 4.9. SCFAs Analyses

As in the previous study, we extracted 10 mg of cecal contents with a mixture of 15% phosphoric acid, diethyl ether, and isocaproic acid. Then, we mixed them well, centrifuged, and removed the supernatant. The supernatant was transferred for analysis through gas chromatography-mass spectrometry (GC-MS) (Agilent Technologies, CA, San Diego, USA). Notably, to construct standard gas chromatographic curves, we procured all standard materials for acetic acid, propionic acid, butyric acid, isobutyric acid, pentanoic acid, isopentyl acid, and hexanoic acid from Sigma Company. Lastly, short-chain fatty acids were calculated quantitatively per the quality of cecal contents and retention time of the standard solution.

### 4.10. Statistical Analysis

All statistical analyses were conducted using SPSS 20.0 and GraphPad Prism. Data are expressed as mean ± SEM. Then, we used Analysis of Variance (ANOVA) to determine the difference in the four groups and Student’s *t*-test to investigate the significance between any two groups. For all experiments, *p* < 0.05 was considered statistically significant.

## 5. Conclusions

This study confirms the protective effect of TBP on DSS-induced colitis mice. TBP treatment significantly alleviates colitis symptoms, including improved body weight and DAI score, inhibited production of pro-inflammatory cytokines, increased antioxidant enzyme activity, weakened colon tissue damage, restored the disturbed intestinal flora, and increased SCFA production. However, further research is needed to understand how TBP plays a role in the gut to produce these effects. In summary, our experimental results will provide some theoretical basis for using TBP as a prebiotic in maintaining intestinal homeostasis.

## Figures and Tables

**Figure 1 marinedrugs-19-00127-f001:**
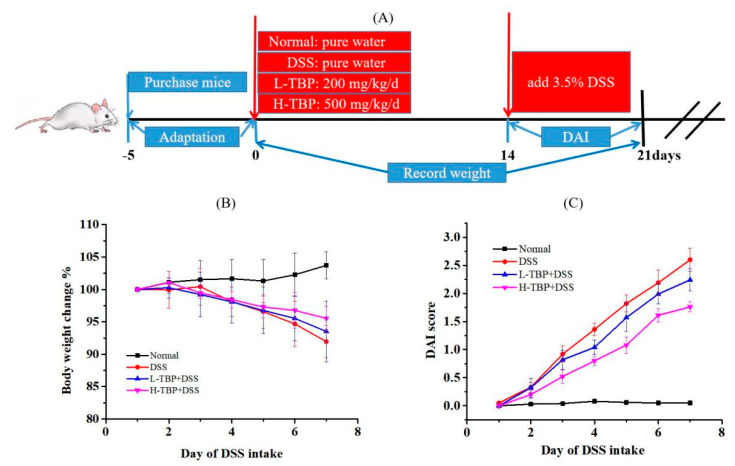
TBP treatment ameliorated dextran sulfate sodium (DSS)-induced colitis mice. (**A**) Diagram of animal experiment, (**B**) the body weight change of each group after DSS-induced, (**C**) disease activity index (DAI) score of each group after DSS was induced. All data are expressed as the mean ± SEM (n = 6).

**Figure 2 marinedrugs-19-00127-f002:**
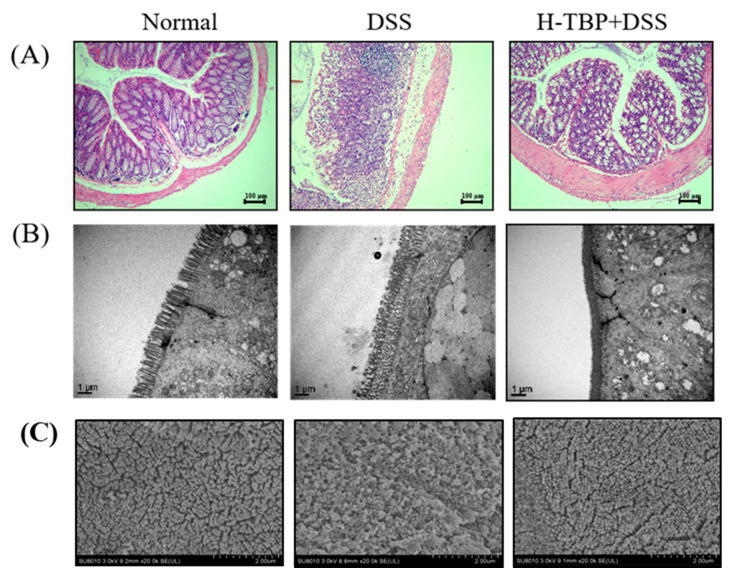
TBP treatment improves DSS-induced intestinal damage. (**A**) Observation of H&E staining of colon tissue sections of mice in each group (100×). (**B**) The colon tissue of each group was observed through TEM (20,000×). (**C**) The colon tissue of each group was observed through SEM (20,000 ×).

**Figure 3 marinedrugs-19-00127-f003:**
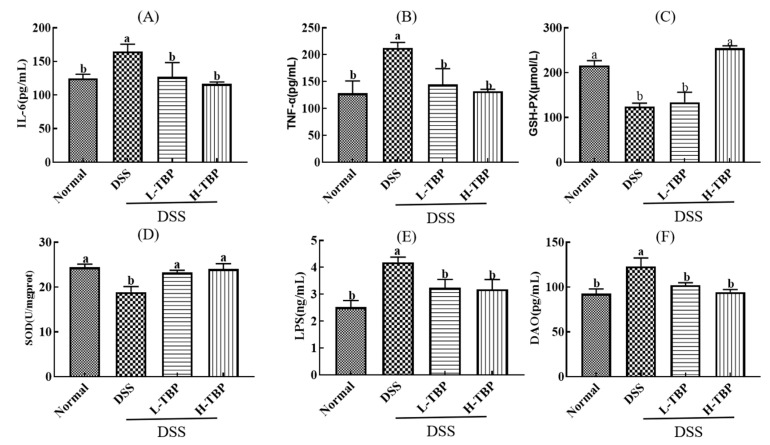
TBP treatment regulates cytokines of colon tissues in DSS-induced colitis mice. (**A**) IL-6, (**B**) TNF-α, (**C**) GSH-PX, (**D**) SOD, (**E**) LPS, (**F**) DAO. Data are presented as the mean ± SEM (n = 3). Different lowercase letters were used to demonstrate significant differences between groups (*p* < 0.05), and the same lowercase letters were used to indicate that the differences between groups were insignificant (P > 0.05).

**Figure 4 marinedrugs-19-00127-f004:**
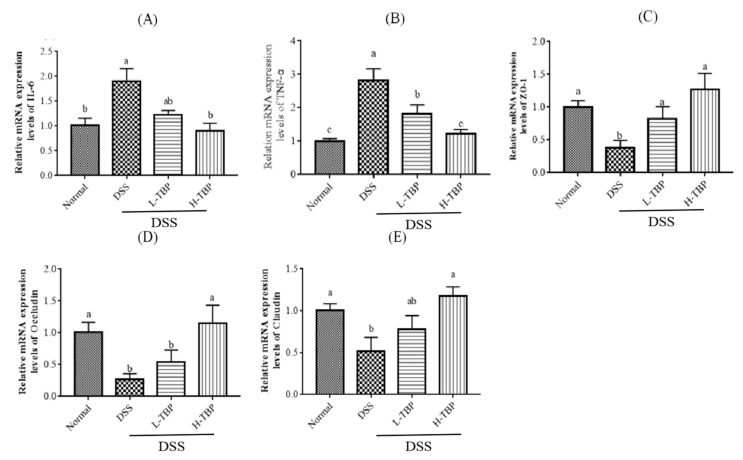
TBP treatment modifies related gene expression in mice colon. (**A**) IL-6, (**B**) TNF-α, (**C**) ZO-1, (**D**) Occludin, (**E**) Claudin. Data are presented as mean ± SEM (n = 3). Different lowercase letters were used to show significant differences between groups (*p* < 0.05), and the same lowercase letters were used to indicate that the differences between groups were insignificant (P > 0.05).

**Figure 5 marinedrugs-19-00127-f005:**
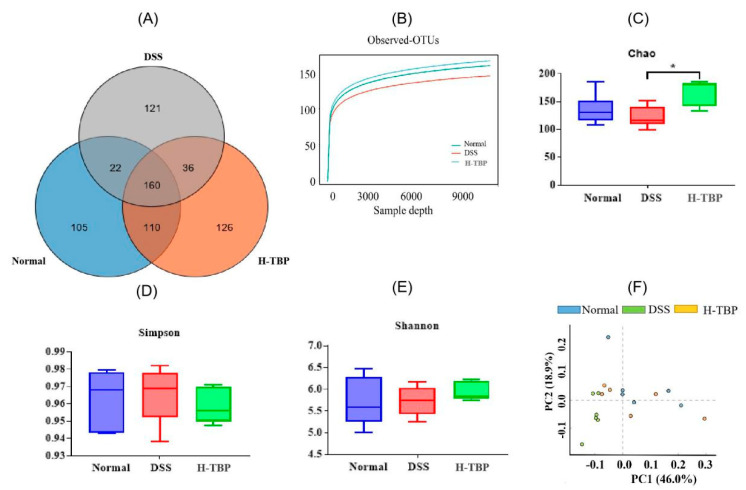
TBP treatment altered the diversity of mice gut microbiota. (**A**)Venn diagrams, (**B**) Observed-OTUs, (**C**) Chao1 index, (**D**) Simpson index, (**E**) Shannon index, (**F**) PCA. All data were expressed as mean ± SEM (n = 6). * means that there is a significant difference between the two groups (*p* < 0.05),

**Figure 6 marinedrugs-19-00127-f006:**
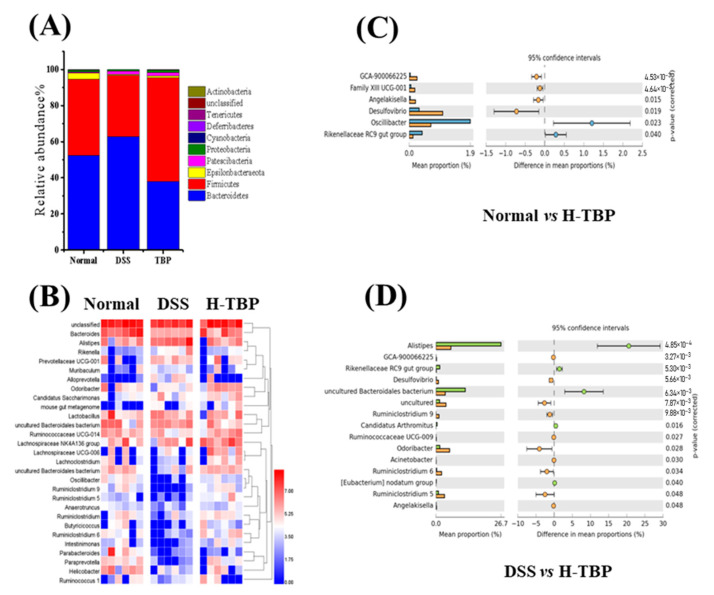
TBP treatment altered the composition of the gut microbiota in mice. (**A**) The community structures of the experimental groups at phylum levels, (**B**) the community structures of the top 20 of the observed sample at genus levels, (**C**,**D**) the gut microbiota had significant differences at genus levels in Normal vs H-TBP, and DSS vs H-TBP. All data are expressed as the mean ± SEM (n = 6).

**Figure 7 marinedrugs-19-00127-f007:**
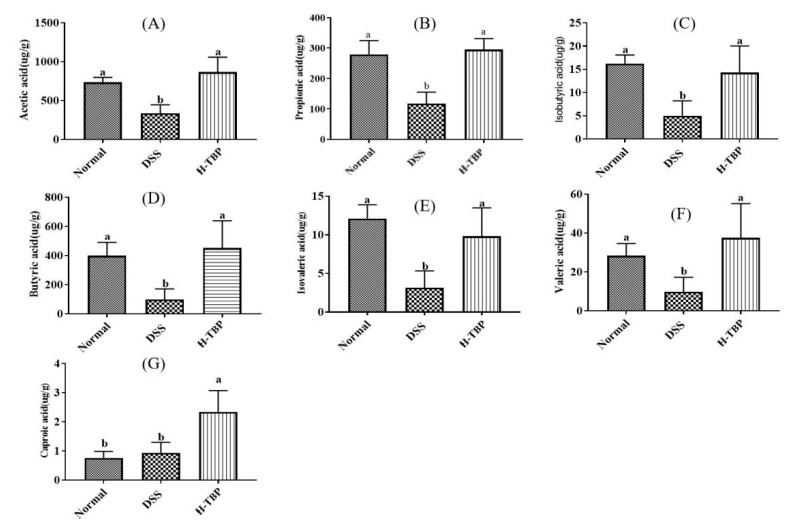
TBP treatment altered the SCFAs in DSS-induced colitis mice. (**A**) Acetic acid, (**B**) Propionic acid, (**C**) Isobutyric acid, (**D**) Butyric acid, (**E**) Isovaleric acid, (**F**) Valeric acid, and (**G**) Caproic acid. Data are presented as the mean ± SEM (n = 6). Means with different letters on bars indicate a significant difference at *p* < 0.05 using ANOVA.

**Figure 8 marinedrugs-19-00127-f008:**
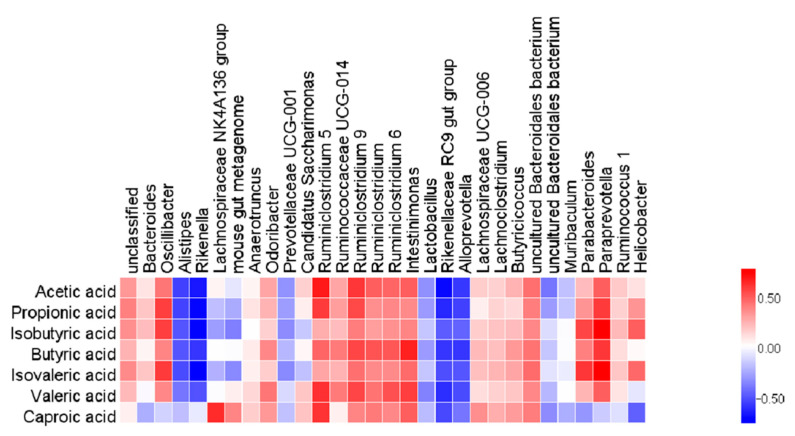
The correlation analysis between gut microbiota and SCFAs. The correlation coefficient R is represented by color, R > 0 illustrates the positive correlation and is characterized by red. R < 0 means a negative correlation, which is shown in blue.

**Table 1 marinedrugs-19-00127-t001:** The molecular weight distribution of tuna bioactive peptides (TBP).

Molecular Weight Range (Da)	Percentage (%)	Molecular Weight Range (Da)	Percentage (%)
>5000	0.16	1000–500	43.23
5000–3000	0.58	500–180	48.71
3000–2000	1.41	< 180	4.61
2000–1000	1.30	-	-
Total		100	

**Table 2 marinedrugs-19-00127-t002:** The amino acid composition of TBP.

Amino Acid	Abbreviation	Ratio (g/100 g)
Glutamic acid ^#^	Glu	9.16
Aspartic acid ^#^	Asp	5.68
Leucine ^*^	Leu	5.35
Lysine ^*^	Lys	5.14
Alanine ^#^	Ala	4.27
Arginine ^#^	Arg	3.67
Valine ^*^	Val	3.42
Histidine ^*^	His	3.09
Glycine ^#^	Gly	3.05
Isoleucine ^*^	Ile	2.96
Proline ^#^	Pro	2.59
Threonine ^*^	Thr	2.58
Serine ^#^	Ser	2.54
Phenylalanine ^*^	Phe	2.43
Methionine ^*^	Met	1.87
Tyrosine ^#^	Tyr	1.78
Cysteine ^#^	Cys	0.52
Essential amino acids ^*^		26.84
Non-essential amino acids ^#^		33.26
Total amino acids		60.10

Notes: An amino acid marked with * means that it is an essential amino acids. An amino acid marked with # means that it is non-essential amino acids.

**Table 3 marinedrugs-19-00127-t003:** Evaluation of disease activity index (DAI).

Weight Loss%	Stool Consistency	Occult/Gross Bleeding	DAI Score
0	Normal	Normal	0
1–5	Soft stools	Hemoccult positive mildly	1
5–10	-	-	2
10–15	Diarrhea	Visible gross bleeding	3
>15	-	-	4

**Table 4 marinedrugs-19-00127-t004:** Primers used for the RT-qPCR.

Gene	Primer	Primer Sequence 5’–3’	Product Size (bp)
IL-6	Forward Reverse	CAGGTCTATTTTGGGATCATTGCC TCCCTGATTTCTAAGTGTTGCTGT	189
*TNF-*α	Forward Reverse	CACCTCAGACAAAATGCTCTTCAC CTCACACATCTCCTTTCTCATTGC	100
ZO-1	Forward Reverse	TACCTCTTGAGCCTTGAACTT CGTGCTGATGTGCCATAATA	248
Occludin	Forward Reverse	GCCCAGGCTTCTGGATCTATGT GGGGATCAACCACACAGTAGTGA	124
Claudin-1	Forward Reverse	GCTGGGTTTCATCCTGGCTTCT CCTGAGCGGTCACGATGTTGTC	110
β-actin	Forward Reverse	AGTGTGACGTTGACATCCGT GCAGCTCAGTAACAGTCCGC	298

## Data Availability

The data presented in this study are available on request from the corresponding author. The data are not publicly available due to public availability violating the consent that was given by research participants.

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
