# Peer review of "Protective Effect of Tuna Bioactive Peptide on Dextran Sulfate Sodium-Induced Colitis in Mice"

_marinedrugs, 2021, doi:10.3390/md19030127_

Round 1

Reviewer 1 Report

In this study, Xiang and colleagues investigated the impact of Tuna Bioactive Peptide on DSS-induced colitis. Although the results indicate extensive work, the data lacks novelty. Various pieces of information and controls are missing making the results difficult to interpret. The text is sloppy and of low quality giving the impression that the authors did not proof-read it before submitting it. There are multiple language and editing issues. The manuscript should be thoroughly corrected before any further consideration.

Major issues

The results lack novelty as multiple different types of bioactive peptides have been shown to have anti-inflammatory properties.

Language

Multiple examples of language and editing issues have been found in the manuscript: grammatical mistakes, interpunction (strongly visible also in the abstract), lacking articles, sentences starting with a small letter, sentences starting with numbers, colloquial vocabulary usually not used in research publications, double spaces between words, lack of space following interpunction signs. The whole manuscript should be thoroughly corrected before considering any submission.

The acronym TBP is introduces 4 times, 1st (Abstract) as "tuna active peptides (TBP)", 2nd line 73 "small molecular weight polypeptide (TBP)", 3rd line 84 "low molecular peptide (TBP)”, 4th  line 306 "low molecular  weight (<1 kDa) peptide components (TBP) ". Funny enough, one of the introduced names fit the acronym TBP!

It is not explained why for some analyses n=3 and others n=6?

Why in some experiments there are 2 TBP groups (L-TBP and H-TBP) in others just one TBP?

Control groups treated with TBP and without DSS are missing.

There is no description of the animal experiment in the results part. To even get to know the experimental group names one has to refer to the materials and methods. Reading would be much easier if the authors add at least one sentence to mention the type of experiment for which they are present results.

The authors supplement the diet and show body weight changes but do not present food intake, diet composition, or energy density of each diet.

Usually, we refer to control groups as “control” and not as “normal”.

Names of the group change between different figures. Some indicate DSS for the TBP group and some do not.

Minor issues:

Abstract

The first word is written in bold font-please correct.

"model group and low- high dose TBP group” this description could be clearer.

"decrease of short-chain fatty acids (SCFAs).“ The word “levels” was omitted in this sentence.

"TBP may play a intestinal protective role in the intestinal tract by inhance the body's antioxidant” Please fix the grammar and avoid repeating words.

Introduction

Line 32 "prevalence rate in China has increased to 11.6/106” What do you mean?

Line 60 "many by-products are produced in the production “ Please fix

Results

Line 86 " The results suggest that TBP may be readily absorbed and utilized in the gastrointestinal tract” What results do the authors refer to? The authors do not present them so I assume that reference is missing here. Besides that, it is a repetition of part of the sentence from lines 81-83…

Line 87-89 "Amino acid analysis showed that TBP contained 18 kinds of amino acids, among which glutamate, aspartate, leucine, alanine and lysine were the main components, which was consistent with previous reports [16].“ How can authors provide a reference for data and then show a table (Table 2) with amino acid composition? There is no reference to Table 2 in the text so it is unknown which data the authors refer to.

Line 90-91 "In addition, the essential amino acids (Val, Leu, Ile, Lys, Phe, Met, Thr, His) accounted for 26.84% of the total amino acids.” No reference, no indication of table or figure.

Table 2 is not well structured. It’s not clear that the sum of AA is listed and the total AA on the bottom. It could be modified to be easier to understand.

Any type of introduction describing the animal experiment is lacking. The reader has to check materials and methods to be able to understand the results.

Reference to Figure 1A in the text is lacking.

Line 95 "Figure 1B showed that the percentage change in body weight of each group mice.” This sentence is not understandable.

Figure 1: The quality of panels B and C is very low. The final time point in panel B lacks error bar lines. Axis title is cropped. Normal is misspelled (“Noraml”) in the figure legends. Acronyms used in the figures are not explained in the legend.

There are no data showing animals' body weight changes upon TBP supplementation (before DSS). The experimental groups in panel A do not fit the groups in panels B and C. There is no explanation of the experimental groups in the text what makes the interpretation of the results impossible.

Figure 2: The background in pictures A is yellow, the scale is not readable. There is no reference to panels A, B, or C in the text.

Figure 3 and 4: What does a, b, and c in the panels mean? What is there the difference between them?

Figure 4: Based on the figures the TBP-treated animals did not get DSS

Figure 8: The legend lacks statistics description and numeric data should be presented as a supplementary table.

Figures 5 and 6: Nearly every panel has a different style and size of fonts. Some of them not readable.

Authors switch between writing (Figure. X) and (Figure X).

Section 2.7. Caproic acid is not SCFA.

Discussion

Summary and outlook missing at the end of the discussion section.

Materials and methods

Table 3: Why there is no description for DAI scores 2 and 4?

Reviewer 2 Report

This paper describes the protective effect of tuna bioactive peptides (TBP) in a mouse model of ulcerative colitis. Although the authors employ many techniques to investigate the activity of TBP and the results are interesting, the paper in its present form is extremely difficult to read and requires extensive re-writing before it is suitable for publication. There is no indication of what type of tuna waste was employed and how the molecular weight and amino acid composition of the hydrolysate was determined. Description of experimental details is very incomplete and does not allow to fully evaluate the results presented by the authors. Most experimental details are completely lacking in the result section. E.g.: there is no definition of what are H-TBP and L-TBP which are shown in Fig. 1B and 1C; there is no indication of how gut microbiota was analyzed to generate the data in fig. 5; it is really difficult to follow the authors' logic if these indications are provided only in the Methods section. Moreover, abbreviations must be defined the first time they are used in the text. 

Other points:

1) why the acronym TBP for tuna active peptides, maybe Tuna Bioactive Peptides?

2) in Table 1 indicate units (Da) in column headings

3) fig. 2B if possible please show images at the same magnification

4) a control group of mice treated with TBP without DSS should have been included to exclude/evaluate direct effects of TBP

Reviewer 3 Report

This manuscript is regarding “Protective effect of tuna bioactive peptide on dextran sulfate sodium-induced colitis in mice". Overall, this manuscript provides valuable findings to this fields. With some major and minor revisions and final proof-reading, this manuscript could be published in Marine drugs.

Major comments:

  1. Line 21: this is not following your result (i.e., decrease of SCFA?)
  2. What is the rationale to feed 200, 500 mg/kg/d of TBP? Authors should give a description regarding how you have come up with these concentrations in the method and/or discussion. For example, based on what responsible component in them?
  3. You have analyzed the composition of amino acids in the TBP. How would you explain whether they (peptide, certain branch of amino acid?) are utilized by microbiota (direct or indirect)? If they do, how are they detected by microbiota? These types of justification/assumption should be deeply discussed in the discussion section.
  4. Figure 4. tight junction related proteins should be measured by their protein expression rather than mRNA expression. 
  5. In general, description regarding the microbiota data is lacking, more detailed explanation is needed for readers.

Minor comments:

  1. Quality of figure should be improved (For example, Figure 1.A, Day 0 and 14: it looks like one time treatment, please consider redraw the design.)
  2. The color of bar in the graph should be consistent throughout the figures. Please revise them accordingly.
  3. Figure legends should have sufficient information that readers could understand your result data just by reading them. Please add detailed information in figure legends including methods.
  4. Please revise figure 3 and figure 4 in a consistent manner.
  5. Figure 7. TBP: is that H-TBP in here since it had superior effect than L-TBP. If so, it should be mentioned somewhere in the text as well as in the figure legend.
  6. Please add limitations of your study in the discussion.

Reviewer 4 Report

The authors show that TBP significantly reduce the mice weight loss; and improve the morphological and pathological characteristics of colon tissue after DSS treatment.

The work is very interesting. The results support the protean effect of these peptides. However, I believe that this work should be completed with some study on the mechanism of action of these TBP. It would be interesting to find out how these peptides act to produce these effects. In addition to some proteins of the tight junctions, it could be studied whether the factor NF-kB or protein kinases related to inflammation and the expression of proteins related to the absorption of sugars and amino acids in the intestine is involved. It would also be interesting to measure the levels of reactive oxygen species in the intestine, not just the activity of antioxidant enzymes.

Minor revision:

In the figures it would be interesting to show the meaning of the abbreviations.

Round 2

Reviewer 2 Report

The manuscript is greatly improved but still lacks clarity in some points. A brief description of the design of the experiment at the beginning of par. 2.2 must be added otherwise the groups presented in fig 1B and 1C remain obscure: H-TBP and L-TBP must be defined in the text not just in the abstract and methods.

Description of preparation of TBP (par. 4.2) is still incomplete: what HPLC column was used? most important, how reproducible is the composition/profile of the TBP obtained by this preparation procedure?

Author Response

Comments 1: The manuscript is greatly improved but still lacks clarity in some points. A brief description of the design of the experiment at the beginning of par. 2.2 must be added otherwise the groups presented in fig 1B and 1C remain obscure: H-TBP and L-TBP must be defined in the text not just in the abstract and methods.

Response: Thank you for your comments. We have added a brief description of the experimental design in par.2.2.

Comments 2: Description of preparation of TBP (par. 4.2) is still incomplete: what HPLC column was used? most important, how reproducible is the composition/profile of the TBP obtained by this preparation procedure?

Response: Thank you for your comments. We have refined the description of TBP preparation in par. 4.2. HPLC (Agilent Technologies, Inc., CA, USA) equipped with TSK gel G2000 SWXL analytical column (7.8×300 mm, 5 µm, TOSOH, Tokyo, Japan) was used to determine the molecular weight distribution of TBP. After several analyses, the composition of TBP has good repeatability under the same extraction method.

Reviewer 4 Report

The paper has been improved and most of the questions asked have been answered and can be accepted in the present form

Author Response

Thank you for your comments.